# Virus-induced transposable element expression up-regulation in human and mouse host cells

Marissa G Macchietto[1], Ryan A Langlois[2], Steven S Shen[1,3]

**Virus–host cell interactions initiate a host cell–defensive response during virus infection. How transposable elements in the host cell respond to viral stress at the molecular level remains largely unclear. By reanalyzing next generation sequencing data sets from dozens of virus infection studies from the Gene Expression Omnibus database, we found that genome-wide transposon expression up-regulation in host cells occurs near antiviral response genes and exists in all studies regardless of virus, species, and host cell tissue types. Some transposons were found to be up-regulated almost immediately upon infection and before increases in virus replication and significant increases in interferon β expression. These findings indicate that transposon up-regulation is a common phenomenon during virus infection in human and mouse and that early up-regulated transposons are part of the first wave response during virus infection.**

## Introduction

Virus–host cell interactions and the spread of virus-derived material initiates host cell–defensive responses during viral infectious disease (Yamauchi & Helenius, 2013). In recent years, many studies in the field have used next generation sequencing technology to assess host cell gene expression profiles, which have led to better understandings of the virus–host cell interaction processes in vitro or in vivo. These studies have provided profound insights into host cell gene expression patterns and antiviral interferon-stimulated gene (ISG) dynamics during infection (Schoggins, 2014; Sjaastad et al, 2018). Most these studies only focused on gene expression profiling and left the activity of transposable elements (TEs) unexplored. TEs, which consist of endogenous retroviruses (ERVs), long interspersed nuclear elements (LINEs), short interspersed nuclear elements, and DNA transposons, are repetitive DNA elements that comprise a large fraction of mammalian genomes (~50%) and have shaped genome structures dramatically over evolutionary time because of their ability to copy (or cut) and paste

to new locations (Chuong et al, 2016; Ito et al, 2017; Platt et al, 2018). Most of the genomic TE sequences have mutated over time to be nonfunctional, and so a single TE is generally unable to produce all the functional proteins required for transposition. However, functionally active TEs that are capable of retrotransposition or combinations of functional pieces of TEs can allow sequence insertions into new locations, which may wreak havoc on normal cellular gene expression if they integrate close to genes or within the coding sequences of genes (Horváth et al, 2017). As a result, aberrant TE expression has been implicated in multiple diseases from cancer to autoimmune disorders (Grandi & Tramontano, 2018).

However, there are also numerous examples of TE sequences being co-opted for use by the host. In mammals, syncytin genes are ERV-derived envelope protein genes which are expressed by cells of the placenta during embryonic development and are involved in placental cell–cell fusion to form the characteristic placental structure (Chuong, 2018). Other placental genes such as corticotropin-releasing hormone have been shown to be regulated by ERV sequence enhancers, showing that ERV sequences are important for placental development and mammalian evolution (Chuong, 2018). Another recent study showed that subsets of ERV and LINE DNA sequences participate in a gene regulatory network that controls innate immune response genes downstream of interferon γ signaling (Chuong et al, 2016), suggesting important roles in host innate immunity.

The mobile ability of TEs make them a source of genomic instability, and thus, during normal cellular activities, they are mostly repressed by numerous epigenetic chromatin modifications (He et al, 2019). However, baseline levels of TE expression exist across different cell types (Tokuyama et al, 2018), and various sources of cellular stress can cause their up-regulation or down-regulation, which may be transient or persistent, and it is unclear how these expression changes affect the cell (Horváth et al, 2017). Several virus infection studies have observed up-regulation of different ERV families because of viruses such as HIV, influenza A virus (IAV), Kaposi's sarcoma herpes virus, and numerous other dsDNA viruses (Ormsby et al, 2012; van der Kuyl, 2012; Chen et al, 2019; Nogalski et al, 2019). However, almost all of these studies have measured ERV mRNA and protein expression through wet laboratory methods and were not able to delve deeper into specific subfamilies and loci that

[1]Institute for Health Informatics, University of Minnesota, Minneapolis, MN, USA  [2]Biochemistry, Molecular Biology and Biophysics Graduate Program, University of Minnesota, Minneapolis, MN, USA  [3]Clinical Translational Science Institute, University of Minnesota, Minneapolis, MN, USA

Correspondence: shens@umn.edu

are affected. To explore the genome-wide TE (specifically ERV and LINE) expression activity during virus infection, we have reanalyzed 37 RNA-sequencing virus infection data sets spanning 25 different types of human and mouse viruses obtained from the Gene Expression Omnibus (GEO) database. We were also able to obtain two mouse and three human virus infection RNA-seq time courses, allowing us to investigate how TEs change over time during the infection process.

We were able to identify 1,715 and 394 differentially expressed TEs (DE TEs) that were shared across several of the human and mouse infection data sets, respectively. These were enriched in the neighborhood of antiviral genes, immune response genes, and the MHC regions in both genomes. The mouse IAV infection time course analysis showed that TE expression changes are some of the earliest transcriptional events during virus infection, with up-regulation occurring even before or concurrent with interferon $\beta$ expression. Together, these results suggest that DE up-regulated TEs may act as a part of an early, conserved host defense response.

## Results

### Genome-wide TE up-regulation during human and mouse virus infections

To investigate genome-wide transposon activity and identify DE TEs, we performed a comprehensive analysis of RNA-seq data from infected and mock-infected cells in both human and mouse. In total, we obtained 37 data sets from the GEO database: 23 human and 14 mouse virus–host data sets with at least two biological replicates per condition, consisting of 19 different human viruses and 8 different mouse viruses (Table S1). We developed a computational pipeline to assess and quantify genes, TEs, introns, as well as transcriptional readthrough (TR) levels, to determine how each changes during infection for each virus–cell pair (Fig 1A).

We used two methods to assess gene and TE expression: 1) TEtranscripts to quantify gene and TE subfamily expression simultaneously for the most accurate measurements of TE expression across treatments (Figs S1 and S2) and 2) featureCounts to leverage uniquely mapped reads in determining the locations of individual TEs that change in expression (Liao et al, 2014; Jin et al, 2015). By comparing DE genes and TE subfamilies quantified with TEtranscripts across virus–host cell data sets (FDR < 0.05), we found numerous genes and TEs that are perturbed by viruses during infection (Tables S2 and S3). We observed TE subfamily up-regulation across the board in different cell types/tissues infected by different viruses in both species, with ERV subfamilies showing the most up-regulation (Fig 2A and B). Viruses that perturbed host gene expression the most also showed the largest changes in expression of TE subfamilies (Fig S3A and B), with ERVs correlating the strongest with gene expression perturbation. These results clearly show that ERV and other TE subfamilies become largely up-regulated during virus infection in different cell types and that the magnitude of their up-regulation (or numbers of up-regulated TEs) is associated with the magnitude of virus-induced gene expression perturbation (or numbers of DE genes) experienced by the host cell. Some viruses did not

elicit large changes in TE expression, such as Kaposi's sarcoma herpes virus, RESTV, and HPV, but they also did not have large effects on gene expression either. We also observed differences in TE up-regulation when the same virus infected different cell types, such as Ebola (EBOV) infecting ARPE-19, monocyte-derived macrophages, and CD4$^+$ T cells. It should be noted that gene and TE expression changes for these data sets may be highly dependent on the time of infection. In addition, viruses interact with host cells via different mechanisms. It is possible that these viruses may be better adapted to circumventing detection by cellular innate immunity factors, causing lesser immune responses in the specific host cell types infected.

Next, we investigated whether certain TE subfamilies and loci are particularly susceptible to virus-induced cellular stress. We found hundreds of TE subfamilies that were up-regulated and shared across ≥3 human and mouse virus data sets (Tables S4 and S5), indicating that particular subfamilies are sensitive to viral stress. Many of these top TE subfamilies constitute internal domains ("-int") of LTR ERVs, which contain protein coding regions required for ERV replication (Figs 1B and S3C). For example, HERVK13-int was up-regulated in 10 human virus data sets, including HIV-infected activated CD4$^+$ T cells (HIV$^+_{activated\ CD4\ T\ cell}$) and resting CD4$^+$ T cells (HIV$^+_{resting\ CD4\ T\ cell}$), IAV, and HSV-1, which have already been shown to express HERVK during infection (Fig 1C) (Kwun et al, 2002; van der Kuyl, 2012). We observed several TE locations in the human and mouse genome that are consistently up-regulated upon infection and are expressed autonomously, which we define here as expression that is independent of gene expression (Fig S4). For the HERVK example in Fig 1C, we can observe that the transcription initiates from a different TE element further upstream, and that their expression is discrete from and not overlapping with neighboring protein coding or long non-coding RNA gene expression. Thus, this would be defined as autonomous TE expression. We performed a Genomic Regions Enrichment of Annotations Tool (GREAT) analysis on 1,715 shared human ERV and LINE loci that become differentially up-regulated across ≥3 virus data sets upon infection, and we found that these TEs are located near genes involved in defense response to virus and cellular response to type I interferon (Figs 1D, S5A, and Table S6) (McLean et al, 2010). Some of the most abundant shared human DE up-regulated TE loci have already been described to shape posttranscriptional regulation of gene expression (L2a/L2c; [Petri et al, 2019]) and contain regulatory motif-binding sites for NF-$\kappa\beta$ and C-rel, which regulate cytokine and proinflammatory genes (L1PA6; [Lawrence, 2009; Macia et al, 2011]) (Fig 1E). We applied the same pipeline to mouse and were able to identify 394 DE TEs (shared ≥3 virus data sets), which were also enriched around genes with similar Gene Ontology (GO) terms related to viral defense response (Fig S5B and C and Table S7). Some of the GO terms also held when the virus data set threshold was dropped and all DE TEs up-regulated in at least one virus data set (47,433 DE TEs) in human (Fig S5D) were included. Last, we wanted to know if any of the human and mouse DE TEs were conserved in the other species. Of the 1,715 human DE TEs, 292 (17%) lifted onto the mouse genome, and two of the lifted TEs were also DE in mouse. For mouse, 44/394 (11%) DE TEs lifted over to the human genome, but none of the TEs in human overlapping these lifted TEs were DE in mouse. These results indicate that TE up-regulation is somewhat conserved across these two species and may be connected to changes in expression of immune response genes.

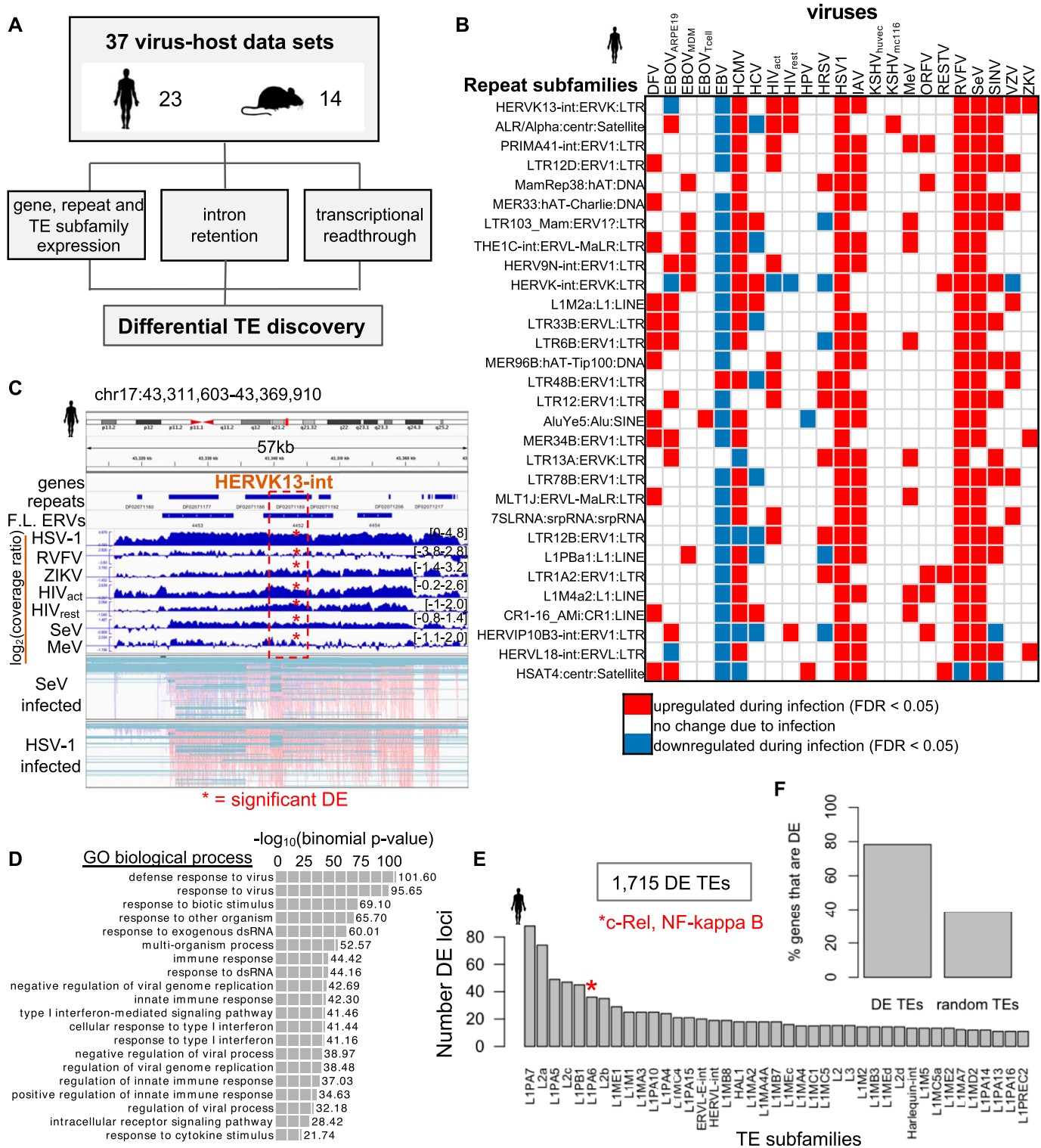

**Figure 1. Transposable elements (TEs) become up-regulated during human and mouse virus infections and are located near antiviral response genes.**
(A) 37 virus–host cell RNA-sequencing data sets in human and mouse were reanalyzed using our analysis pipeline. Gene, repeat element, repeat subfamily expression, intron retention, and transcriptional readthrough (TR) were quantified for each virus data set. (B) Top 30 shared differentially expressed (DE) TEtranscripts repeat subfamilies shared across human virus data sets. (C) Genome viewer screenshot of an HERVK13-int subfamily locus that is one of the top shared DE up-regulated subfamilies during human virus infections. HERVK13-int is a full-length endogenous retrovirus (ERV) element indicated by "F.L ERVs" track (Tokuyama et al, 2018). Virus coverage tracks show the log2 ratio between the Reads Per Kilobase (of gene exon) per Million reads mapped (RPKM)-normalized coverage of infected over mock samples. Bottom two tracks show read alignments in HSV-1 and SeV infected samples. (D) Genomic Regions Enrichment of

We observed a correlation between TE and gene expression changes, so next, we questioned if TE expression is linked to the expression of their adjacent genes. We associated TEs in the human genome with their nearest gene neighbors and found that 78% of the genes that are closest to the 1,715 of up-regulated DE TEs are also DE, whereas only 39% (95% confidence interval [38.96%, 39.10%], 1,000 randomizations) of genes are DE when equal numbers of TEs are chosen at random (Fig 1F). This indicates that there is a connection between DE TEs and their nearest DE genes. Next, to inspect these relationships further, we looked at DE TEs and genes in each virus data set using only DE TEs that appears in the shared set. We determined the DE status of every gene nearest to each DE up-regulated TE in each virus data set that were also within the shared DE TE set (1,715 DE TEs that are present in ≥3 VDs) for human. Grouping non-DE genes with genes that change in the opposite direction of DE up-regulated TEs, we find that slightly more human virus data sets have genes that are not DE or DE in the opposite direction of their neighboring DE TEs (12 data sets) than human virus data sets that have genes that are DE in the same direction as their neighboring DE TEs (nine data sets) (Fig S6A). For mouse, nine (60%) virus data sets showed more DE genes expressing in the same direction as their neighboring DE up-regulated TEs, whereas only six (40%) virus data sets showed more DE genes in the opposite direction and/or were not DE (Fig S6B). These results indicate that a sizeable portion of DE TEs are likely connected to DE gene expression.

## Relationship between DE TEs and other transcriptional events

Some viral proteins have the ability to perturb host gene splicing and transcriptional termination, resulting in intron retention (IR) and TR of host gene transcripts (Hennig et al, 2018; Chauhan et al, 2019). To further examine how TE expression is impacted by their proximity to genes, we quantified IR and TR in each virus data set. To quantify IR, we compared reads overlapping gene introns across mock and infected samples. Gene introns were considered DE if the cumulative reads mapping into introns were DE, and the gene was either not DE or DE in the opposite direction of the intron (Fig S7A). This ensured that introns are not becoming DE as a result of significant changes in gene expression. We quantified the numbers of genes with IR across the virus data sets and found increased IR in hundreds to thousands of genes in infected samples. Cases of IR also correlated strongly (r = 0.7) with the numbers of genes perturbed because of infection and genes with IR were shared across human and mouse virus data sets (Fig S7B–D), indicating that transcript splicing issues are prevalent in host cells during virus infection and that TE sequences within introns may derive their expression from IR.

TR is a common occurrence across the human genome, but cellular stress (e.g., osmotic, oxidative, and heat stress) has been shown to cause substantial increases in the number and length of gene transcripts with readthrough (Vilborg et al, 2017). To determine how different virus infections change gene readthrough, we implemented DoGFinder, a tool designed to identify genomic regions with uniform coverage downstream of genes, into our pipeline (Vilborg et al, 2015). We found differences in the lengths of readthrough regions as a result of virus infection (Fig S8A and B), with lengths of TR generally increasing. Our findings confirmed observations of TR in HSV-1 and IAV, which were reported by other studies (Heinz et al, 2018; Hennig et al, 2018). Generally, TR was observed in data sets where gene expression perturbation was high. In addition, we witnessed numerous instances in the Genome Browser where TR continued into neighboring genes, creating an IR phenotype in the neighboring gene and boosting the perceived neighboring gene's expression (Fig S8C).

To determine if TE up-regulation is primarily due to gene-related transcriptional events or is expressed autonomously, or independently of gene expression, we overlapped up-regulated DE TE loci with annotated TRs and other gene regions. We found that DE TEs originate from TR (data set range: 0–39%), IR (32–88%), upstream regions (3 kb from TSS) (0.4–10%), downstream regions (3 kb from TTS, but is not annotated as TR) (5–31%), and intergenic regions (4–37%) (Fig 3A). However, when we view these annotated TEs in the Genome Browser, we observed that these relationships between TEs and genes are more complex; there are regions with clear intergenic and autonomous TE expression (Figs 3B and C and S4A), intron regions that have clear autonomous TE expression (Fig S4B), intron regions that have TE expression from IR, TR regions that yield TE expression (Fig 3D), and we also observed cases where some IR and intergenic TE expression was in fact because of TR of neighboring genes (Fig S8C). We manually inspected and annotated the top 88 shared up-regulated TE loci across human viruses using the Genome Browser. We found that 9%, 19%, and 40% of the inspected TEs are clearly intergenic, from IR, or from TR, respectively, and the remaining 32% are difficult to discern (Table S8). Manual annotation of the top 60 TEs in mouse showed 28%, 32%, and 23% from intergenic regions, IR, and TR, respectively (Table S9). Thus, we found that DE TE expression originates from multiple sources, including intergenic regions, IR, and TR in response to virus infections, and the number of autonomous DE TEs in this report is a conservative estimate because it is difficult to ascertain the exact breakdown of the TE sources using the current short-read data and computational methods.

Next, we overlapped the 1,715 human DE TEs and the 394 mouse DE TEs with FANTOM 5 CAGE data to determine what fraction of the identified DE TEs have been previously shown to initiate transcription from their sequences in other data sets and under different conditions (Lizio et al, 2015). We found that 79/1,715 (4.6%) human and 59/394 (15%) mouse DE TEs have evidence of transcription initiation in FANTOM 5 and thus are likely autonomously transcribed TEs during virus infection. We inspected the locations of the 79 human and 59 mouse DE TEs that overlap CAGE regions and interestingly found that the majority are found within gene introns (54% in human and 58% in mouse) followed by downstream regions (24% in human and 24% in mouse) (Fig S7E and F).

Annotations Tool (GREAT) GO terms for genes that are located proximal to 1,715 DE TEs that are shared by ≥3 human virus data sets. **(E)** Identities of some of top TE subfamilies based on occurrence of shared human DE TE loci. **(F)** Bar chart showing the fraction of DE genes (≥3 human virus data sets) that are proximal to 1,715 DE TEs (≥3 human virus data sets) in comparison with genes proximal to 1,715 random TEs.

**A** (human)

**Number of DE subfamilies**

| Virus | Cell type | Group | GEO study | ERV Up | ERV Down | LINE Up | LINE Down | SINE Up | SINE Down | DNA Up | DNA Down | tRNA Up | tRNA Down | other Up | other Down | total DE genes | total DE repeats |
|---|---|---|---|---|---|---|---|---|---|---|---|---|---|---|---|---|---|
| HSV-1 | HFF-1 | dsDNA | GSE100576 | 558 | 0 | 170 | 0 | 58 | 0 | 283 | 0 | 33 | 0 | 56 | 0 | 4458 | 1158 |
| EBV | CD19+ B cells | dsDNA | GSE126379 | 3 | 450 | 1 | 152 | 0 | 56 | 1 | 241 | 1 | 3 | 1 | 37 | 10372 | 946 |
| HCMV | MRC5 | dsDNA | GSE120890 | 274 | 32 | 129 | 2 | 41 | 3 | 178 | 13 | 0 | 0 | 23 | 14 | 3876 | 709 |
| VZV | MeWo | dsDNA | GSE85493 | 25 | 16 | 16 | 4 | 5 | 0 | 4 | 3 | 0 | 0 | 2 | 0 | 3119 | 75 |
| HPV | HFK | dsDNA | GSE92496 | 0 | 6 | 0 | 1 | 0 | 4 | 0 | 11 | 5 | 0 | 3 | 6 | 3001 | 36 |
| ORFV | HFF-1 | dsDNA | GSE93226 | 24 | 1 | 4 | 0 | 0 | 0 | 4 | 1 | 0 | 0 | 1 | 0 | 2757 | 35 |
| KSHV | MC116 | dsDNA | GSE119608 | 3 | 0 | 0 | 0 | 0 | 0 | 0 | 0 | 1 | 0 | 1 | 3 | 1240 | 8 |
| KSHV | HUVEC | dsDNA | GSE119608 | 0 | 0 | 0 | 0 | 0 | 0 | 0 | 0 | 2 | 0 | 0 | 0 | 55 | 2 |
| DFV | huh-7.5 | (+) ssRNA | GSE110512 | 64 | 24 | 29 | 6 | 13 | 1 | 60 | 5 | 0 | 2 | 10 | 5 | 4139 | 219 |
| HCV | huh-7.5 | (+) ssRNA | GSE103730 | 16 | 85 | 3 | 28 | 0 | 2 | 6 | 11 | 0 | 0 | 2 | 7 | 7651 | 160 |
| SINV | HEK293 | (+) ssRNA | GSE125182 | 69 | 6 | 10 | 1 | 1 | 1 | 12 | 10 | 0 | 0 | 7 | 2 | 392 | 119 |
| ZIKV | MDM | (+) ssRNA | GSE118305 | 25 | 1 | 2 | 0 | 0 | 0 | 2 | 1 | 2 | 0 | 4 | 1 | 2022 | 38 |
| HIV | activated CD4+ T cells | (+) ssRNA | GSE122735 | 30 | 9 | 0 | 0 | 1 | 0 | 3 | 3 | 1 | 0 | 3 | 3 | 4030 | 53 |
| HIV | resting CD4+ T cells | (+) ssRNA | GSE122735 | 6 | 1 | 0 | 1 | 1 | 0 | 0 | 0 | 0 | 0 | 1 | 1 | 3588 | 11 |
| RVFV | HSAEC | (-) ssRNA | GSE102481 | 518 | 1 | 163 | 1 | 58 | 0 | 265 | 0 | 27 | 0 | 49 | 1 | 8001 | 1083 |
| SeV | Namalwa B cells | (-) ssRNA | GSE115266 | 332 | 68 | 127 | 12 | 42 | 0 | 208 | 19 | 24 | 2 | 32 | 10 | 10704 | 876 |
| IAV | A549 | (-) ssRNA | GSE82232 | 215 | 0 | 54 | 0 | 51 | 0 | 44 | 0 | 41 | 0 | 30 | 1 | 3964 | 436 |
| HRSV | A549 | (-) ssRNA | GSE99298 | 27 | 52 | 0 | 26 | 3 | 0 | 8 | 18 | 2 | 7 | 1 | 10 | 6542 | 154 |
| EBOV | ARPE-19 | (-) ssRNA | GSE105414 | 80 | 16 | 3 | 2 | 0 | 0 | 11 | 1 | 5 | 0 | 6 | 3 | 6847 | 127 |
| MeV | glioma cells | (-) ssRNA | GSE111247 | 47 | 11 | 6 | 1 | 1 | 0 | 21 | 4 | 2 | 0 | 4 | 1 | 1138 | 98 |
| EBOV | MDM | (-) ssRNA | GSE84188 | 10 | 14 | 3 | 1 | 2 | 0 | 2 | 1 | 0 | 0 | 1 | 2 | 2796 | 36 |
| EBOV | CD4+ T cells | (-) ssRNA | GSE99389 | 3 | 0 | 0 | 0 | 2 | 0 | 0 | 0 | 1 | 0 | 0 | 1 | 719 | 7 |
| RESTV | MDM | (-) ssRNA | GSE84188 | 4 | 0 | 0 | 0 | 0 | 0 | 0 | 0 | 0 | 0 | 1 | 0 | 384 | 5 |

Left-side category labels (A): dsDNA / retrovirus (+) ssRNA / (-) ssRNA

**B** (mouse)

| Virus | Cell type | Group | GEO study | ERV Up | ERV Down | LINE Up | LINE Down | SINE Up | SINE Down | DNA Up | DNA Down | tRNA Up | tRNA Down | other Up | other Down | total DE genes | total DE repeats |
|---|---|---|---|---|---|---|---|---|---|---|---|---|---|---|---|---|---|
| HSV-1 | LIM | dsDNA | GSE74215 | 233 | 10 | 78 | 2 | 21 | 0 | 26 | 4 | 0 | 0 | 4 | 2 | 8906 | 380 |
| MCMV | NK cells | dsDNA | GSE113214 | 139 | 19 | 87 | 1 | 31 | 0 | 34 | 3 | 0 | 0 | 7 | 0 | 6477 | 321 |
| MHV68 | NIH 3T3 | dsDNA | GSE70481 | 15 | 2 | 3 | 0 | 0 | 0 | 5 | 0 | 0 | 0 | 2 | 0 | 821 | 27 |
| HSV-1 | DRG | dsDNA | GSE74215 | 1 | 1 | 0 | 0 | 0 | 0 | 0 | 0 | 0 | 0 | 0 | 0 | 639 | 2 |
| WNV | BMDM_48hpi | (+) ssRNA | GSE104817 | 144 | 2 | 59 | 0 | 2 | 0 | 6 | 0 | 0 | 0 | 2 | 0 | 4579 | 215 |
| WNV | BMDM_12hpi | (+) ssRNA | GSE104817 | 111 | 7 | 56 | 0 | 14 | 0 | 8 | 0 | 0 | 0 | 2 | 3 | 2379 | 201 |
| MCV | 17Cl-1 | (+) ssRNA | ERP013565 | 26 | 5 | 0 | 0 | 0 | 0 | 7 | 3 | 0 | 0 | 4 | 1 | 2939 | 46 |
| IBV | ciliated lung epithelial cell | (-) ssRNA | GSE115952 | 86 | 221 | 41 | 54 | 17 | 10 | 15 | 38 | 0 | 1 | 13 | 5 | 13534 | 501 |
| IAV | AM | (-) ssRNA | GSE115904 | 264 | 0 | 114 | 0 | 32 | 0 | 29 | 0 | 0 | 0 | 11 | 0 | 1889 | 450 |
| IAV | AECII | (-) ssRNA | GSE115904 | 132 | 1 | 39 | 0 | 21 | 0 | 5 | 1 | 0 | 0 | 6 | 0 | 4159 | 205 |
| H5N1 | lung | (-) ssRNA | SRP061303 | 143 | 28 | 40 | 4 | 1 | 1 | 6 | 9 | 0 | 0 | 4 | 2 | 7060 | 238 |
| LCMV | NK cells | (-) ssRNA | GSE113980 | 80 | 15 | 14 | 3 | 0 | 0 | 4 | 3 | 4 | 0 | 2 | 3 | 3923 | 128 |
| H5N8 | lung | (-) ssRNA | SRP061303 | 29 | 1 | 0 | 0 | 0 | 0 | 0 | 1 | 0 | 0 | 1 | 0 | 769 | 32 |
| IAV | CC | (-) ssRNA | GSE115904 | 11 | 1 | 0 | 0 | 0 | 0 | 0 | 0 | 0 | 0 | 1 | 0 | 690 | 13 |
| IAV | CD103 | (-) ssRNA | GSE115904 | 7 | 0 | 0 | 0 | 0 | 0 | 0 | 0 | 0 | 0 | 0 | 0 | 352 | 7 |

Left-side category labels (B): dsDNA / (+) ssRNA / (-) ssRNA

Legend:
down / up — # DE repeat subfamilies:
0 to 5
6 to 10
11 to 20
21 to 60
61 to 100
101+

low — high

**Figure 2. Genome-wide repeat subfamily changes during viral infection.**
**(A, B)** Number of ERV, long interspersed nuclear element (LINE), short interspersed nuclear element (SINE), DNA, tRNAs, and other repeat subfamilies that are differentially up- and down-regulated during virus infection in human (A) and mouse (B) from TEtranscripts. Viruses are categorized by type of genetic material, and host cell types and Gene Expression Omnibus (GEO) study IDs are indicated. Total numbers of DE genes and DE repeat subfamilies are included in the far right columns, and values are scaled from high (red) to low (white) for each virus genetic material category.

## Dynamics of TE up-regulation during virus infection

To determine the dynamics of TE up-regulation during virus infection, we turned to a published 7-d influenza A time course (GSE49933) of mouse lung tissue containing 20 time points (0, 3–4, 7–8, 11–12, 26–28, 32, 49–50, 74–75, 98–99, 122–123, and 148–149 h) (Altboum et al, 2014). This experiment measured gene expression from the 3′ end of transcripts and not from the full-length of transcripts (Table S1). We plotted the numbers of DE TEs, numbers of DE genes, and virus and interferon β expression levels in counts per million (CPM), each normalized by their totals, over the time course (Fig 4A). We observed that interferon β expression levels tightly correlated with levels of virus transcription in the tissue, and although interferon β only becomes significantly up-regulated by 26 hours post-infection (hpi), we

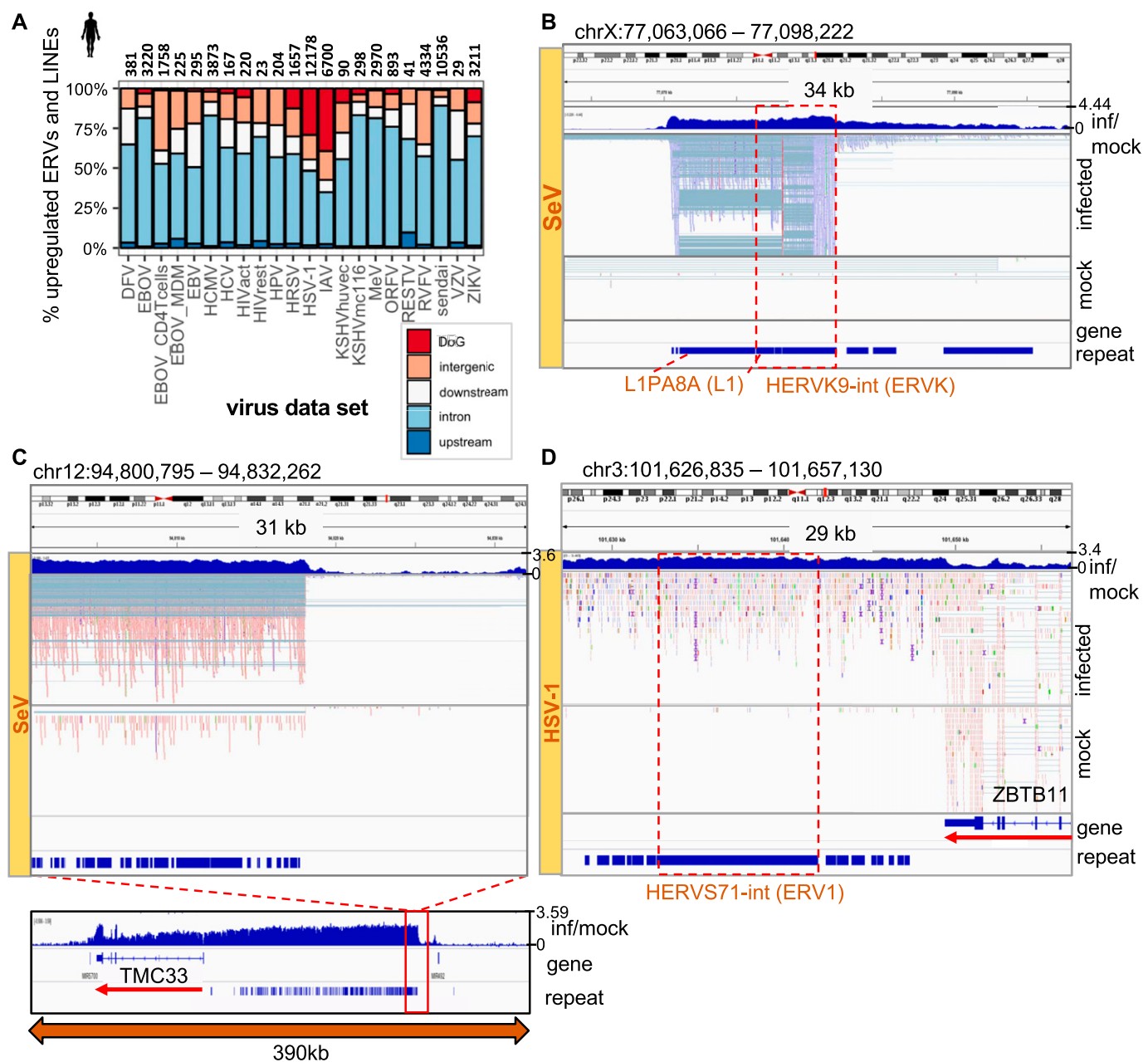

**Figure 3. Relationship between differentially up-regulated TEs and other transcriptional events.**
**(A)** Significantly up-regulated TEs and their relationship with genes across human virus data sets. Total numbers are indicated above each bar. Upstream and downstream regions are defined as 3 kb from the transcriptional start site or termination site, respectively. Intergenic refers to an element that is greater than 3 kb from any annotated ENSEMBL gene boundaries. **(B, C, D)** Genome browser view showing (B) intergenic TEs that are highly up-regulated in SeV-infected cells, (C) up-regulation of a stretch of hundreds of TE and repeat elements ~100 kb upstream of TMC33 gene during SeV infection, and (D) TR of the ZBTB11 gene into HERVS71-int (ERV1) in HSV-1-infected cells. "inf/mock" track shows the log₂(RPKM-normalized coverage of infected over mock samples). "repeat" and "gene" tracks show DFAM repeat and RefSeq gene model annotations, respectively.

observed that several TEs become significantly up-regulated as early as 3 hpi. More than this, whereas the % DE genes peak after virus expression peaks, % DE TEs peak before virus, interferon-$\beta$, and % of DE genes, indicating that TEs are among the earliest responding genomic elements to IAV infection.

We divided the time course into early (0–26 hpi), middle (49–75 hpi), and late (99–150 hpi) stages of infection based on the viral expression profile and performed a GREAT analysis on the DE TEs up-regulated in each stage. We found that these up-regulated DE TEs lie around genes related to interferon $\beta$ response and innate immune response and observed GO terms that were unique and specific to early up-regulated TEs such as response to biotic stimulus, response to other organism, and regulation of multi-organism process (Fig 4B).

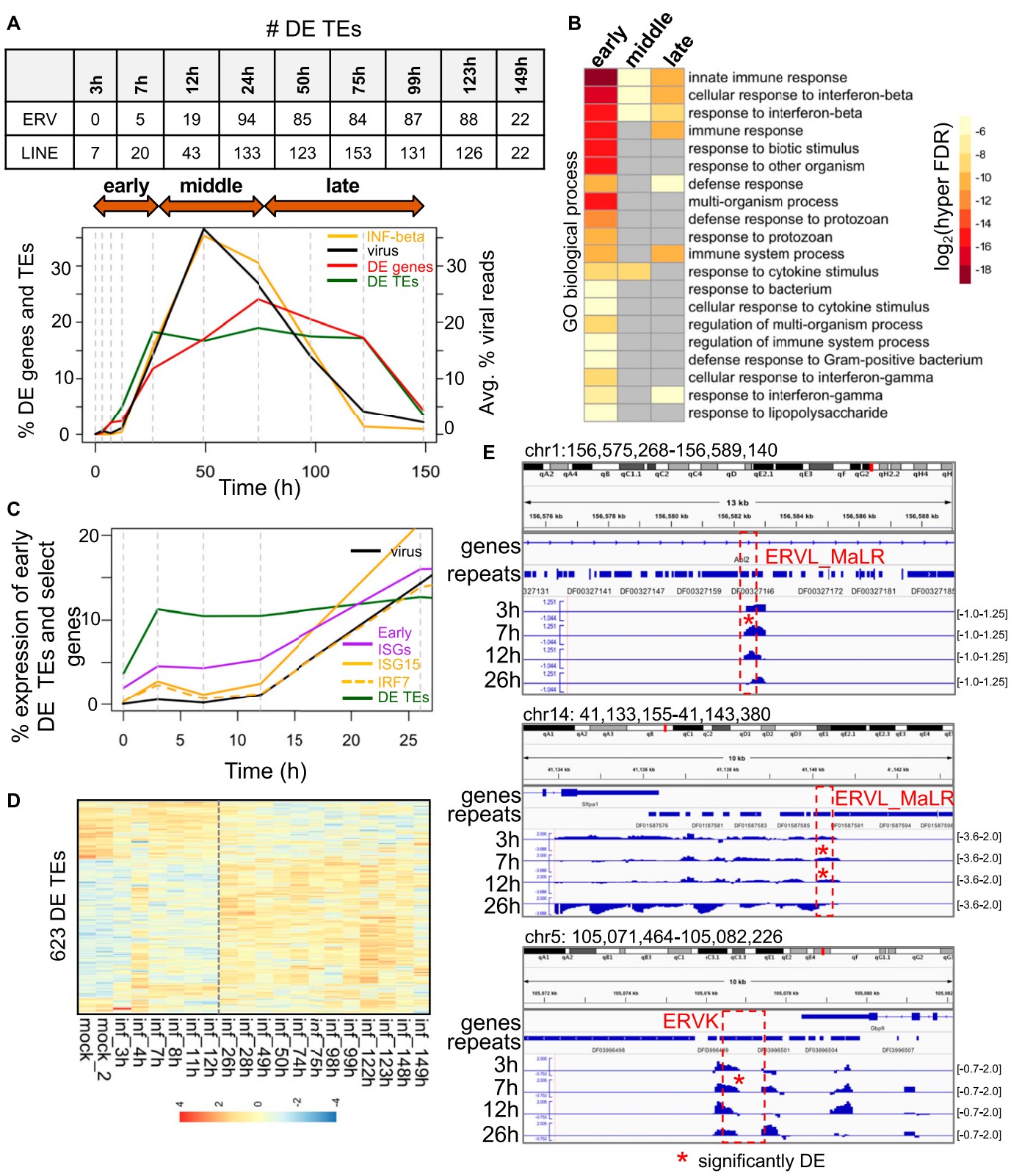

**Figure 4. TEs are up-regulated very early during influenza A virus (IAV) infection, peaking before significant changes in IFN-β gene expression.**
**(A)** Percentages of DE genes (up and down-regulated), DE TEs (up and down-regulated), IFN-β expression (percent of CPM expression), and viral reads (% reads corresponding to IAV in each sample) during a 7-d mouse IAV time course, scaled relative to the total over the time course. Vertical dashed grey lines indicate the time points. The table above the plot shows the number of DE ERV and DE LINE elements at each time point. **(B)** Biological process GO terms corresponding to genes proximal to DE TEs that are expressed in the early, middle, and late stages of infection. **(C)** Percentage of early DE TEs (up at 3, 7, and 12 h), early interferon-stimulated genes (ISGs)

Next, we looked at how overall early up-regulated DE TE expression changed over the time course. We plotted the sum of early (3, 7, and 12 h) DE TE expression, early (3, 7, and 12 h) gene expression, and early (3, 7, and 12 h) DE ISG expression in CPM relative to the sum of their expression changes over the time course. We also plotted the viral gene expression and the expression of an interferon stimulated gene, *Isg15*, and *Irf7*, a regulatory factor that activates interferon and interferon response genes (Fig 4C). The expression profile of *Isg15* closely mirrored the interferon β expression profile, but unlike interferon β, *Isg15* showed significant up-regulation as early as 3 hpi, whereas *Irf7* showed significant up-regulation at 12 hpi. This indicates that ISG expression is changing significantly before significant changes in interferon β expression, and this may likely be due to the presence of low levels of interferon protein. Several LINEs and short interspersed nuclear elements that were significantly up-regulated at 3 hpi were found within intergenic regions and gene introns (Fig 4E), whereas ERVs were found to be significantly up-regulated at 7 hpi and within intergenic regions, downstream regions, and introns. However, of 61 early up-regulated ERVs and LINEs (up by 12 h), none were associated with or in close proximity to *Isg15*, and only two TEs were associated with other ISGs (*Stat1* and *Akt3*) indicating that these expression changes were likely not due to changes in ISG expression. TE numbers were mostly sustained beyond 26 hpi and did not continue to increase even as the number of DE genes increased (Fig 4A and D). TE up-regulation before substantial gene expression changes also suggests that these early up-regulated TEs are autonomous and unaffected by other transcriptional events. We found 623 TE elements changing in expression during the time course (Fig 4D). We overlapped these 623 TEs and 241 early up-regulated TEs (within 26 h of infection) with FANTOM 5 CAGE data and found 44/623 (7.1%) and 16/241 (6.6%) overlap FANTOM transcription initiation peaks, respectively.

We validated DE TE dynamics with the data obtained from four virus strains from two independent studies: a 20-h mouse norovirus (MNV) infection time course in mouse RAW 264.7 cells (GSE96586, Fig S9) (Levenson et al, 2018) and a 24-h IAV time course comparing IAV (H3N2) Brisbane, Udorn, and Perth strains in a human cell line (GSE61517) (Fabozzi et al, 2018) (Fig S10). We witnessed dramatic changes in numbers of TEs changing expression by the first time point across all time courses (Figs S9A–F and S10A–C). Although TE up-regulation before overall gene expression changes is not as obvious in these time courses as in the 7-d IAV time course because of a lack of early time points, we were able to observe DE TEs as early as 6 h during the human IAV time course and statistically significant up-regulation of TEs, including some autonomous ERVs before significant changes in interferon-β expression, which also became DE at 24 h (Fig S10A–C and Table S10). Thus, these time courses provide evidence to show that up-regulation of TEs is still a very early event during mouse and human virus infections, that they are sensitive to virus-induced stress, and are able to change their expression before significant changes in interferon expression.

## Virus-induced DE TEs are enriched in the MHC class region

Unlike its sister regions MHC I and II that have clearly defined functions in immune response and antigen presentation, the MHC III region is less defined structurally and functionally but is known to be related to physiological stress responses and inflammation (Yu et al, 2000; Janeway et al, 2001). It contains genes such as heat shock proteins, tumor necrosis factor, and complement component genes of the innate immune system (Yu et al, 2000). We investigated this region because an early up-regulated ERV during the IAV time course mapped to a mouse RefSeq-annotated viral envelope gene (D17H6S56E-5) 200 kb upstream of a tumor necrosis factor gene within this region. The expression of this ERV is increased at 3 hpi (FC = 1.56, *P*-value = 0.003, FDR = 0.70; not significant) and is statistically significantly differentially up-regulated by 7 hpi. Other mouse virus infections showed significant up-regulation as well, indicating that this locus is very sensitive to virus infection (Fig 5A). We suspected that this ERV envelope protein may also become up-regulated in response to general cellular stress due to its early timing. To determine that, we inspected its expression in 3T3 fibroblasts subjected to heat stress, oxidative stress ($H_2O_2$), and osmotic stress (KCl) from a TR study (Vilborg et al, 2017) (Fig 5A). Surprisingly, although these stress conditions cause substantial up-regulation of TEs in general and some baseline level of expression exists at this ERV locus in the 3T3 fibroblasts, we found that this viral envelope gene was not significantly up-regulated by any of these stress conditions, further indicating that this is a specific response to virus-induced cellular stress. In addition, we compared all 7,075 DE TE loci that were found in ≥1 mouse virus data set to all 8,347 TE loci that were DE in ≥1 nonvirus stress condition. We found minimal (6%) overlap between them (Fig S11), although we observed a fairly decent degree of overlap (38%) between the viral (7,584 DE genes shared by ≥3 mouse virus data sets) and nonviral (7,399 DE genes shared by ≥1 nonvirus stress data set) sources of stress. This indicates that the vast majority of TEs that are up-regulated during viral infection are unique to viral stress, at least when compared with this limited set of nonviral stress conditions.

Next, we questioned if there is a connection between early DE TEs and the MHC regions. We performed a Fisher's exact test on numbers of DE TEs in each MHC region during the IAV time course and found that MHC class III was significantly enriched with DE TEs beginning at 12 h during the IAV time course when compared with the genomic background or to the other MHC regions (Fig 5B and C). When performed for other mouse virus infections, we found 71%, 57%, and 25% of the viral data sets had significant enrichment in MHC class III, class I, and class II, respectively. In human, we saw that 4%, 56%, and 43% of virus data sets have TE enrichment in MHC III, I, and II, respectively, suggesting that TEs in MHC III region in human are not as sensitive to viral stress. Using the human IAV time course data, we inspected the up-regulated TE enrichment in these regions. For the human Udorn IAV time course, we observed TE

(up at 3, 7, and 12 h; purple line), *Isg15* (solid orange line), *Irf7* (orange dashed line), and virus expression (CPM) over the time course, normalized by their expression totals over the entire time course. **(D)** Heat map showing row-scaled, $\log_2$ (CPM expression) of 623 DE TEs that are up-regulated or down-regulated at some point during the IAV time course. **(E)** Genome viewer screenshots showing early DE ERV elements that are expressed within introns or downstream of genes. Time point tracks show $\log_2$(RPKM expression ratio of infected over mock-infected samples). The y-axis range is shown on the right.

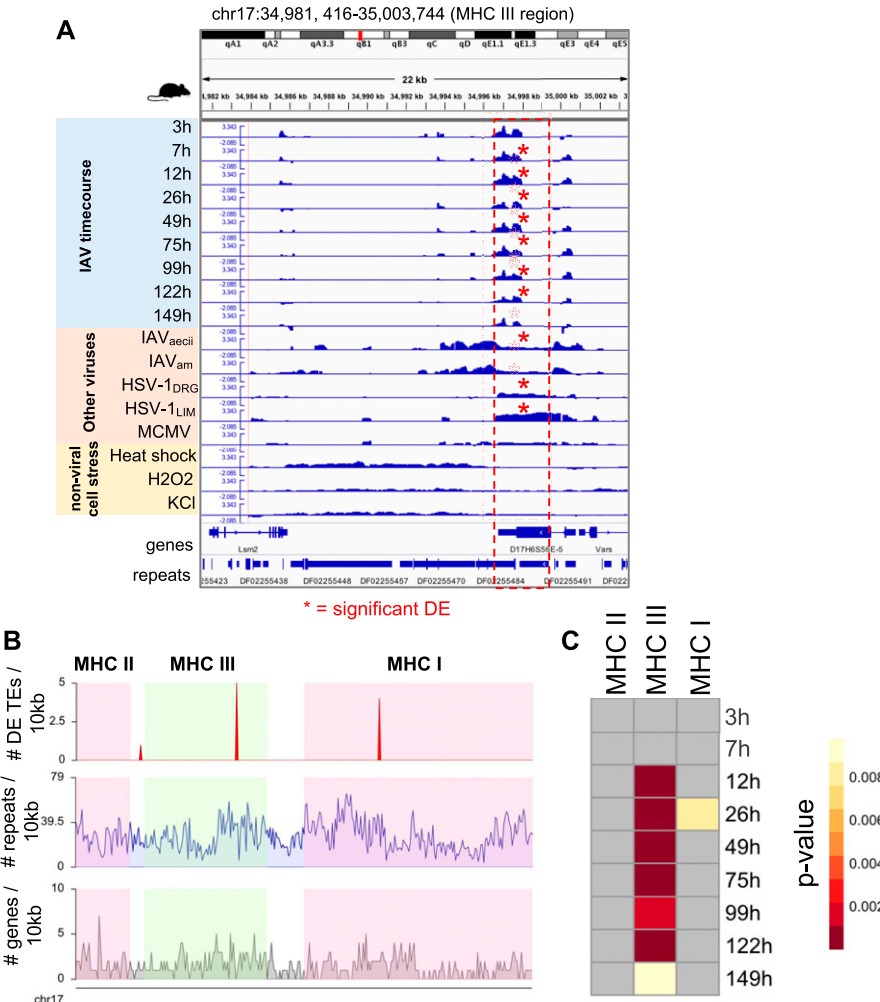

**Figure 5. Early up-regulated TEs are enriched in the MHC III region during IAV infection time course.**
**(A)** Genome viewer screenshot showing early up-regulation (12 hpi) of a RefSeq-annotated viral envelope protein located in MHC III and shared up-regulation with other mouse virus data sets. Each data set track shows bam-normalized coverage (RPKM, log ratio) of stressed samples (viral and nonviral stress) to unstressed samples. Red asterisks indicate DE; IAV time points are DE if FDR < 0.25, whereas other data sets are DE if FDR < 0.05. Y-axis for all tracks is −2.085–3.343. **(B)** Gene, repeat, and DE TE density in the MHC region of mouse (mm10) chromosome 17: 33860353-37090447. Pink and green shaded regions indicate the boundaries of the MHC regions selected for performing a Fisher's exact test based on MHC gene annotations from Shiina et al (2017). **(C)** Heat map showing TE enrichment in MHC regions during the IAV time course. Grey cells have *P*-values of 1 (no TE enrichment).

enrichment in MHC I, II, and III at 6 and 24 hpi. The human IAV Perth strain time course showed up-regulation of MHC I and II at 6 h, and the IAV Brisbane time course showed up-regulation of MHC II at 6 h and MHC I at 24 h, indicating that there are differences in TE enrichment across the MHC regions that appear to be due to slight genetic differences in these viruses (Table S11). These results demonstrate that virus up-regulated TEs are can be significantly enriched in MHC regions during the early stages of infection, and that slight genetic differences across viruses can impact the degree of TE up-regulation seen in these and other regions.

## Discussion

Although many studies have examined the genome-wide response of host cells to viral infection at the level of the host's genes, TEs have not been very well characterized during these interactions. Host ERVs, which are the remnants of ancient retroviruses, and LINEs comprise a large fraction of mammalian genomes and play important roles in host gene regulation, including of those genes involved in the innate immune response (Chuong et al, 2016). Up to this point, several studies have discovered that there are increased levels of ERV RNAs and/or proteins in different virus infections in different cells (Kwun et al, 2002; Ormsby et al, 2012; van der Kuyl, 2012; Dai et al, 2018), but there have been no in-depth characterizations of TE subfamilies and loci during these viral infections. Moreover, it has been unclear if these events occur in response to all virus infections or if it is specific to virus or tissue types, and whether the TE expression up-regulation is connected to known perturbations of gene expression that occur during viral infection. Ultimately, is TE expression the byproduct of virus-induced dysregulation of gene expression that serves no purpose or is it part of a conserved host cell defense response?

In this study, we have used comprehensive computational tools to explore the genome-wide expression activity of TEs from published data sets that were originally designed for gene expression profiling of human and mouse virus infections. The results have demonstrated that genome-wide TE up-regulation occurs in host cells of both mouse and human during virus infections, and that it is a common phenomenon. We observed that cell type, virus, and time of infection are all biasing factors that will influence the degree of

TE up-regulation, but overall, almost all data sets showed more TE up-regulation after infection than down-regulation. Only one data set showed dramatically opposite patterns to the general trend we observed for TE up-regulation, IR, and readthrough: EBV-infected CD19[+] B cells in latency stage III. EBV-infected CD19[+] B cells showed massive down-regulation of TEs, as well as decreased IR and readthrough. We believe that this data set has opposite behavior because of either virus- and host-specific interactions or to the specific laboratory conditions in which the samples were prepared. Because EBV is able to transform B-cells, it is possible that the TE down-regulation we are observing may be due more to a changing cell state than to an infection response. HIV-infected activated and resting T-cells are another example of this; activated T-cells which is a more differentiated form of the resting T-cells showed dramatically lower levels of TEs than resting T-cells in mock samples, and also had fewer TEs changing during HIV infection than resting T cells. Thus, the TE up-regulation trends still generally hold across virus infections. In addition, beyond virus-specific and host cell differences, each virus experiment was collected at different infection time points, which will also influence the degree of TE up-regulation observed as we have already seen from multiple virus infection time courses.

We observed several intergenic full-length ERVs that became up-regulated and were shared across several human and mouse virus data sets, showing that particular loci are sensitive to virus-induced cellular stress. Moreover, we observed that up-regulated TE loci appear near genes involved in antiviral defense, interferon $\beta$ response genes, and the MHC region in both species with a small number of these TEs showing conservation between species. We also observed minimal overlap of virus-induced TEs with TEs up-regulated by nonviral sources of stress (6%), which may suggest that some TE loci in the mouse genome are specifically sensitive to virus infection stress, although more nonviral stress data sets will need to be analyzed for comparison.

We witnessed autonomous TE expression as well as gene-related expression through IR and TR events as a result of virus infection. We posited that if gene expression were influencing TE detection, we would expect to see up-regulated TEs only associated with up-regulated genes. Instead, we observed that the top up-regulated TEs are associated with a mixture of DE up-regulated, DE down-regulated, and non-DE genes. However, it is possible that genes may not change expression, but their intron expression and TR will change, thereby influencing TEs within those regions. These TE up-regulation cases should be classified as linked to gene expression, but given the used experimental designs and computational methods, these cases cannot be reliably teased apart. To ascertain the exact transcriptional associations between TEs and genes, full-length transcriptome sequencing will need to be performed in the future. In addition, our study does not directly investigate the canonical activity of TEs, such as insertion and translocation events, during viral infection stress. However, multiple studies in plants, yeast, and *Drosophila* have shown that cellular stress does change canonical TE activity (Kunze et al, 1997; Grandbastien, 1998; Magwire et al, 2011; Guio et al, 2014; Mateo et al, 2014), and so it is plausible that viral infection stress can change canonical TE activity as well and should be investigated in future studies.

However, our analyses and manual annotations of some of the top DE TEs have shown that DE TEs detected by our pipeline are products of both gene-related transcriptional events and gene-

independent events (~10%). Regardless of where these TE sequences are originating from, transcribed TE sequences are capable of forming dsRNAs that can be recognized by pattern recognition receptors (PRRs) (such as RIG-1 and MDA5), which can trigger a host immune response. Although laboratory verification is still needed to support the connection between the DE TEs we have observed and host cell gene expression regulation, these results provide some valuable evidence and a catalog of DE TEs that can be used for perturbation testing in the future.

Using several time course studies to determine the timing of TE up-regulation, we observed for the first time that many TEs are up-regulated very early during infection (within 3 h). More strikingly, in the IAV time course study, some TEs are up-regulated before increases in virus replication and significant interferon-$\beta$ gene expression (DE at 26 h) but did show concurrent up-regulation with an interferon stimulated gene, *Isg15* (DE at 3 h). This indicates that low levels of interferon protein are able to stimulate ISG expression. Closer inspection of early DE TE expression showed a large increase by 3 hpi, and then expression dips during the middle infection when interferon and virus levels are highest and increases again when interferon levels decrease towards the end of infection. We witnessed TE up-regulation occurring before DE interferon $\beta$ expression in five time courses in two species from three independent studies, showing that these events are likely not specific to a single virus infection but is somewhat shared across viruses, cell types, and species. However, we did witness significant up-regulation of interferon genes such as Isg15 and Irf7 before interferon-$\beta$, as early as 3 and 12 h, respectively, indicating that low levels of interferon $\beta$ transcript and protein may be enough to dramatically alter interferon response gene expression. Last, we observed that different TEs are turned on during early infection than later infection and early DE TEs appear to be more associated with viral defense genes than later DE TEs.

Based on this work, we propose a new virus–host interaction model (Fig 6) that is an extension of Schoggin's model (Schoggins,

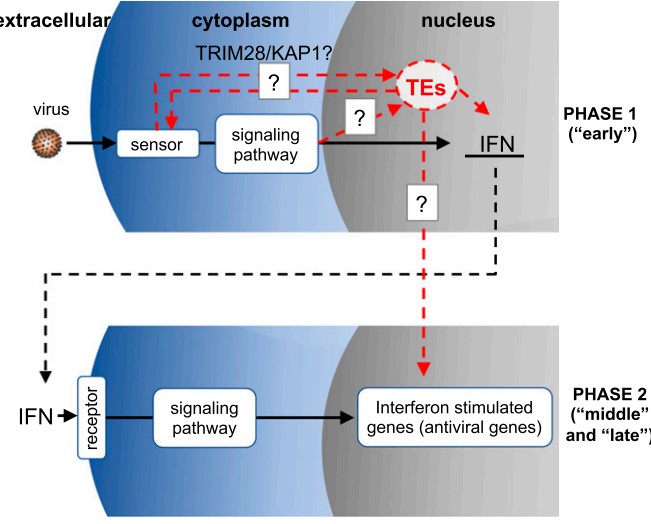

**Figure 6. Model of TE up-regulation during early infection.**
Phase 1: host cell senses virus via different mechanisms and triggers transcription of IFN and TEs Red dashed line corresponds to "early" phase of IAV time course. Phase 2: IFN triggers the transcription of antiviral ISGs.

2014), where the host can sense virions docking on or entering through the membrane and subsequently up-regulate both TEs and interferon. One possibility is TE mRNAs and proteins, especially full-length ones, could be further activating the innate immune response by triggering PRRs, producing a more robust response during early infection (Schneider et al, 2014; Hurst & Magiorkinis, 2015). Interestingly, a very recent study found that influenza A and B viral infection stress reduce levels of SUMOylated (small ubiquitin-like modifications) TRIM28/KAP1 protein, which is a major epigenetic repressor of ERVs, resulting in up-regulation of immunostimulatory ERV sequences and up-regulation of innate immune response genes (Schmidt et al, 2019). Using TRIM28 KO cells together with a TRIM28 construct that cannot be sumoylated, they showed that ERV and innate immune response gene signatures were enhanced during viral infection of the modified TRIM28 relative to wild-type TRIM28. This study proposes that TRIM28 SUMOylation acts as a regulatory switch in the host viral defense response by up-regulating ERVs that will trigger cellular PRRs (like RIG-1 and MAVS) via dsRNAs, further up-regulating interferon and ISGs (Schmidt et al, 2019). In addition, they showed that loss of SUMOylated TRIM28 was independent of PRRs and interferon α (Schmidt et al, 2019), supporting what we have seen in the expression time courses where TEs become up-regulated very early or during infection before significant changes in antiviral response genes. Thus, this work not only supports the work that we have found, but it may be also the primary mechanism behind how TE up-regulation is achieved across these virus infections. In conclusion, this work supports genome-wide TE involvement in viral stress, shows TE dynamics during infection, and provides evidence to suggest that it may be part of a conserved host defense response.

## Materials and Methods

### Virus–host RNA-sequencing data

Virus data sets in Table S1 were acquired from NCBI's GEO or the Short Read Archive database. Only virus data sets containing multiple (2+) replicates of uninfected/mock-infected cells and virus-infected cells were used for the analysis. For all data sets, only the first read pair of paired-end read data sets was used. This was performed to provide consistency across data sets because some data sets used only single-end reads and some paired-end data sets had low-quality coverage of the second read pair. The human genome version hg38 and the mouse genome version mm10 from ENSEMBL with their corresponding ENSEMBL annotations (GRCh38.91 for human and GRCm38.90 for mouse) were used for this analysis. Repeat annotations for both human and mouse were downloaded from the RepeatMasker Web site (version 4.0.6; http://repeatmasker.org) using the DFAM 2.0 database (hg38-Dec 2013-RepeatMasker-open-4.0.6-Dfam 2.0 & mm10-Dec 2011-RepeatMasker open-4.0.6-Dfam 2.0) (Hubley et al, 2015). Virus genomes and annotations were downloaded from NCBI, and the virus genome accessions are provided in Table S1.

### Read alignment

Virus–host RNA-seq data sets were aligned to the individual species' genomes or a combination of the individual species genomes and the specific virus genome using the STAR aligner (Dobin et al, 2013) with the following settings: −outFilterMultimapNmax 100 −winAnchorMultimapNmax 100. Large numbers of multi-mapped reads were allowed so that they could be quantified by TEtranscripts later.

### TE subfamily and gene expression quantification

To determine the TE subfamily and gene expression simultaneously, we used TEtranscripts from the TEToolKit (Jin et al, 2015). TEtranscripts is designed to accurately quantify TE subfamilies and genes, by collapsing multi-mapping reads associated with repeat subfamilies. Mouse and human DFAM repeat annotations were converted into special GTF files that are compatible with TEtranscripts using a custom in-house script. TEtranscripts was run in using −mode multi, -n TC, and the unstranded setting was applied to all data sets. Virus percentages in each sample for each data set were determined from the TEtranscripts output. However, before differential expression analysis, virus reads were removed from the counts matrix. DE repeat subfamilies and genes were determined using edgeR (Robinson et al, 2010), and a significance threshold FDR < 0.05 was applied to genes and TE subfamilies (scripts in GitHub). DE TE subfamilies and genes that were shared across data sets were determined using an in-house script ("shared_DErepeats.py") from edgeR results files.

### Quantification of repeat loci

To determine TE loci that are DE between mock and infected cells, we quantified counts mapping to TE and gene features separately with featureCounts (Liao et al, 2014) from the Subread package counting only uniquely mapping reads. TE features that overlapped gene exons were filtered out from the annotation file before quantification using bedtools (Quinlan, 2014). Simple repeat elements and viral genes were also removed. The TE and gene count matrices were combined to call DE gene and TE features in edgeR using a threshold of FDR < 0.05 (Robinson et al, 2010).

To compare relative coverage of infected and mock-infected samples, biological replicate bam files were merged, and their coverages were compared using bamCompare from deepTools 3.1.1 (Ramírez et al, 2014) with the following settings: −operation log2 −extendReads 300 −effectiveGenomeSize 2308125349 -of bigwig -p 10 −scaleFactorsMethod None −normalizeUsing RPKM −ignoreDuplicates −pseudocount 1.

### Intron retention

To quantify intron retention, we determined the coordinates of all transcript introns from the human and mouse gtf files. Portions of intron regions that overlap exons were removed, leaving only coordinates that correspond to intronic sequence only. Introns were added back to the gene GTF file with exon feature IDs and gene attribute IDs containing the information of the gene that they are derived from, along with the suffix "_intron." Gene and gene introns were quantified using featureCounts, and DE genes and introns were determined with edgeR using different significance thresholds for genes (FDR < 0.05 and $\log_2$(FC) > 2X) and introns (FDR < 0.05 and $\log_2$(FC) > 1X) (Robinson et al, 2010). To make sure that intron

expression changes were not due to changes in gene expression, we filtered out DE introns that are expressed in the same direction as its corresponding gene, if the gene was also significantly DE using a custom script. Thus, we only retained DE introns where their corresponding gene is not DE during infection or is DE in the opposite direction of the intron.

### TR

To determine TR regions, we used DoGFinder (Vilborg et al, 2015), a tool that is designed to identify and quantify readthrough transcription by searching for uniform coverage past annotated gene 3′ transcriptional stop sites using a sliding window. Refseq and ENSEMBL annotations were combined to form a comprehensive list of all possible 3′ transcriptional stop sites for each species. Read counts were normalized to equal depths after removal of gene counts and TR candidates were identified using the following (less stringent) parameters: -minDoGLen 1000, -minDoGCov 0.5, -w 200, and -mode F. DoG regions can span up until the boundary of the neighboring gene, prohibiting us from identifying the actual DoG lengths and associating intronic TEs with DoGs of upstream genes.

### Associating DE TEs with DE genes

To determine if there is an association between the DE TEs that are up-regulated across multiple human virus data sets and genes that they are nearby, we took all 1,715 DE TEs (ERVs and LINEs) ≥3 human virus data sets, and determined their closest gene neighbors using bedtools closest (v 2.25.0). We calculated the percentage of gene neighbors that are DE across at least one data set. As a control, we selected equal numbers of randomly selected ERVs and LINEs from the genome (1,000 iterations) and determined how many DE genes that they are nearby.

### Associating DE TEs with TR regions

To determine if DE TEs are originating from TR regions in each virus data set, repeat annotations filtered of repeats that overlap gene exons (-a) were overlapped with the union of annotated mock and infected DoGs (union DoGs; -b) from each virus data set with bedtools intersect with -wo setting. A custom in-house script categorized and quantified DE up-regulated TEs for each virus data set based on whether they overlap DoGs or not. If they did not overlap annotated DoG regions, they were further subdivided into categories such as intergenic, in introns, within 3 kb downstream of genes, or within 3 kb upstream of genes. Downstream regions were included here because we observed cases of signal downstream of genes that did not meet the standards of a DoG region with DoGFinder. In cases where a TE has multiple relationships with genes, we categorized using the following hierarchical scheme: intron > downstream > upstream.

To determine if DE TEs are enriched in DoG regions, we compared DE TEs in DoGs and non-DoG regions with non-DE TEs in DoGs and non-DoG regions using a Fisher's exact test.

### Associating DE TEs with transcription initiation peaks from FANTOM 5

Shared up-regulated human and mouse DE TEs were overlapped with a comprehensive set of FANTOM 5 (October 2017) CAGE sites that represent transcriptional initiation sites found across available cell lines/tissues. We used bedtools intersect (v 2.25.0) to identify TEs from these sets whose coordinates intersect the CAGE peak coordinates by a single base pair.

### GREAT analysis

To determine if significantly differentially up-regulated repeats during infection are associated with any particular annotations, genomic coordinates for 1,715 individual LINE and ERV elements that are DE up-regulated in two or more human virus data sets were lifted over to the hg19 genome using the UCSC genome browser liftover tool using default settings. The features that lifted over were run through GREAT using the UCSC hg19 genome assembly. Repeat coordinates were associated with the genes whose basal regulatory domain is defined as 5 kb upstream and 1 kb downstream, plus an extended ("distal") regulatory domain up to 1,000 kb. The whole genome was used as background.

### 7-d influenza, 20-h norovirus, and 24-h IAV strain infection time course analysis

The 7-d influenza A (GSE49933), 20-h norovirus time course (GSE96586), and three 24-h influenza A (H3N2) strain time courses (Brisbane, Udorn, and Perth; GSE61517) were downloaded from GEO and mapped and quantified identically to other mouse and human virus data sets (Altboum et al, 2014; Levenson et al, 2018). The 7-d IAV time course did not have biological replicates for each time point, so time points that were 1–2 h apart were grouped together as biological replicates for DE analysis using a less stringent significance cutoff (FDR < 0.25) for gene and TE-calling against the uninfected (0 h) time point. For the norovirus time course, an FDR of 0.05 was applied. The 24-h IAV time courses had mock-infected sample time points to complement each infection time point. Thus, each infection time point was compared with the mock-infected time points instead of 0 h, and significant DE genes, repeats, and TEs were called using an FDR threshold of 0.05.

To examine the relationships between changing DE genes and TEs, the numbers of DE genes and TEs at each time point were divided by the total number of DE genes and TEs over the time courses, respectively. Expression of interferon-$\beta$ was overlayed on the top after first dividing CPM expression of each time point by the total expression during the time course.

To produce the MNV time course of TE-gene heat map (Fig S9F), TEs were associated with their closest ENSEMBL gene neighbors using bedtools closest. The expression (in CPM) of TEs that became DE during the 20-h time course were plotted side by side to the expression of the closest gene neighbor during the time course.

### Data access

The data sets analyzed during the present study are available in the GEO repository, and the study accessions and file accessions are

provided in Table S1 and also in the main text. Additional code is available on GitHub (https://github.com/mmacchietto/TE_virus_project).

## Supplementary Information

## Acknowledgements

We thank Dr Ashley Haase and Dr David Masopust from the University of Minnesota for insightful discussions. This work is supported by UL1 TR002494.

### Author Contributions

MG Macchietto: data curation, software, formal analysis, validation, and methodology.
RA Langlois: resources and data curation.
SS Shen: conceptualization, resources, and investigation.

### Conflict of Interest Statement

The authors declare that they have no conflict of interest.

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
