## [Reviewer comments · Life Science Alliance]

Life Science Alliance

Virus-induced transposable element expression upregulation in human and mouse host cells

Marissa Macchietto, Ryan Langlois, and Steven Shen

DOI: <https://doi.org/10.26508/lsa.201900536>

Corresponding author(s): Steven Shen, University of Minnesota, Institute for Health Informatics, Clinical Translational Science Institute

Review Timeline:

Submission Date:	2019-08-27
Editorial Decision:	2019-10-01
Revision Received:	2019-12-05
Editorial Decision:	2020-01-03
Revision Received:	2020-01-09
Accepted:	2020-01-10

Scientific Editor: Andrea Leibfried

Transaction Report:

No Peer Review Process File is available with this article, as the authors have chosen not to make the review process public in this case.

October 1, 2019

Re: Life Science Alliance manuscript #LSA-2019-00536-T

Steven S Shen

University of Minnesota, Institute for Health Informatics, Clinical Translational Science Institute

Dear Dr. Shen,

Thank you for submitting your manuscript entitled "Virus-induced transposable element activation in human and mouse host cells" to Life Science Alliance. The manuscript was assessed by expert reviewers, whose comments are appended to this letter.

As you will see, the reviewers appreciate your analysis, but raise some concerns with the data/their interpretation that need to get addressed to allow publication here. All three reviewers provide constructive input on how to revise your work and we would thus like to invite you to submit such a revised version to us. Most requests can get addressed by re-analysis of existing data, still requiring major revision efforts. The reviewers also point out that your work refers to TE activation without providing data supporting this. Addressing this point with data would clearly strengthen the value of your work to others tremendously, so we would like to encourage you to include such data (eg via comparing RNA-seq and CAGE-seq data if available).

Thank you for this interesting contribution to Life Science Alliance. We are looking forward to receiving your revised manuscript.

Sincerely,

Andrea Leibfried, PhD
Executive Editor
Life Science Alliance
Meyershofstr. 1
69117 Heidelberg, Germany
t +49 6221 8891 502
e a.leibfried@life-science-alliance.org
www.life-science-alliance.org

B. MANUSCRIPT ORGANIZATION AND FORMATTING:

January 3, 2020

RE: Life Science Alliance Manuscript #LSA-2019-00536-TR

Prof. Steven S Shen
University of Minnesota, Institute for Health Informatics, Clinical Translational Science Institute
516 Delaware St. S.E.,
Minneapolis, MN 55455

Dear Dr. Shen,

Thank you for submitting your revised manuscript entitled "Virus-induced transposable element expression upregulation in human and mouse host cells". As you will see, the reviewers now support publication, and we would thus be happy to publish your paper in Life Science Alliance pending final minor revisions:

- Please address reviewer #2's concern by stating in the relevant section of the manuscript the rationale for using only the first read pair of paired-end reads
- Please upload all figure files, including suppl. Figures, as individual files and without figure legends; the figure legends should all go into the main manuscript docx file, please
- Please add callouts to Fig S7D, S9A-E, and S10 to the manuscript text

A. FINAL FILES:

-- Summary blurb (enter in submission system): A short text summarizing in a single sentence the study (max. 200 characters including spaces). This text is used in conjunction with the titles of papers, hence should be informative and complementary to the title. It should describe the context

and significance of the findings for a general readership; it should be written in the present tense and refer to the work in the third person. Author names should not be mentioned.

B. MANUSCRIPT ORGANIZATION AND FORMATTING:

Sincerely,

Andrea Leibfried, PhD
Executive Editor
Life Science Alliance
Meyershofstr. 1
69117 Heidelberg, Germany
t +49 6221 8891 502
e a.leibfried@life-science-alliance.org
www.life-science-alliance.org

January 10, 2020

RE: Life Science Alliance Manuscript #LSA-2019-00536-TRR

Prof. Steven S Shen
University of Minnesota, Institute for Health Informatics, Clinical Translational Science Institute
516 Delaware St. S.E.,
Minneapolis, MN 55455

Dear Dr. Shen,

Thank you for submitting your Research Article entitled "Virus-induced transposable element expression upregulation in human and mouse host cells". It is a pleasure to let you know that your manuscript is now accepted for publication in Life Science Alliance. Congratulations on this interesting work.

DISTRIBUTION OF MATERIALS:

Again, congratulations on a very nice paper. I hope you found the review process to be constructive and are pleased with how the manuscript was handled editorially. We look forward to future exciting submissions from your lab.

Sincerely,
